# Brief communication: Comparing hydrological and hydrogeomorphic paradigms for global flood hazard mapping

Giuliano Di Baldassarre[1,2,3], Fernando Nardi[4,5,6], Antonio Annis[4], Vincent Odongo[1,3], Maria Rusca[1,3], and Salvatore Grimaldi[7]

[1]Department of Earth Sciences, Uppsala University, Uppsala, Sweden
[2]IHE-Delft Institute for Water Education, Delft, The Netherlands
[3]Centre of Natural Hazards and Disaster Science, CNDS, Sweden
[4]WARREDOC, University for Foreigners of Perugia, Perugia, Italy
[5]Institute of Environment, Florida International University, Miami, USA
[6]Fondazione Eni Enrico Mattei (FEEM), Milan, Italy
[7]Tuscia University, Viterbo, Italy

*Correspondence to*: Giuliano Di Baldassarre (giuliano.dibaldassarre@geo.uu.se)

**Abstract.** Global floodplain mapping has rapidly progressed over the past few years. Different methods have been proposed to identify areas prone to river flooding, resulting into a plethora of available products. Here we assess the potential and limitations of two main paradigms, and provide guidance on the use of these global products in assessing flood risk in data-poor regions.

## 1 Premise

As economic losses and fatalities caused by river flooding have dramatically increased over the past decades (Winsemius et al., 2016), there has been much progress in the development of analytical tools for the identification of the areas that can be potentially flooded (Ward et al., 2015; Dottori et al., 2018; Nardi et al., 2019). This progress has also been accelerated by the adoption of the Sendai Framework for Disaster Risk Reduction and the Warsaw International Mechanism for Loss and Damage Associated with Climate Change Impacts (Ward et al., 2015). As such, more and more scientists, experts and practitioners use global floodplain maps in data-poor regions for the identification of flood risk hotspots or the mapping of flood-prone areas (Ward et al., 2015; Winsemius et al., 2016; Dottori et al., 2018; Nardi et al., 2019).

## 2 Hydrological mapping

There are two main paradigms to map flooding. The traditional paradigm is (implicitly or explicitly) based on a definition of the floodplain as the area falling within the extent of a given flood event. In this *hydrological* paradigm, a range of synthetic event with a given probability of occurrence or return period (Pappenberger et al., 2013; Ward et al., 2015; Dottori et al., 2018), such as the 1-in-200 year flood event, is typically estimated via hydrological modelling or statistical analysis of flood data.

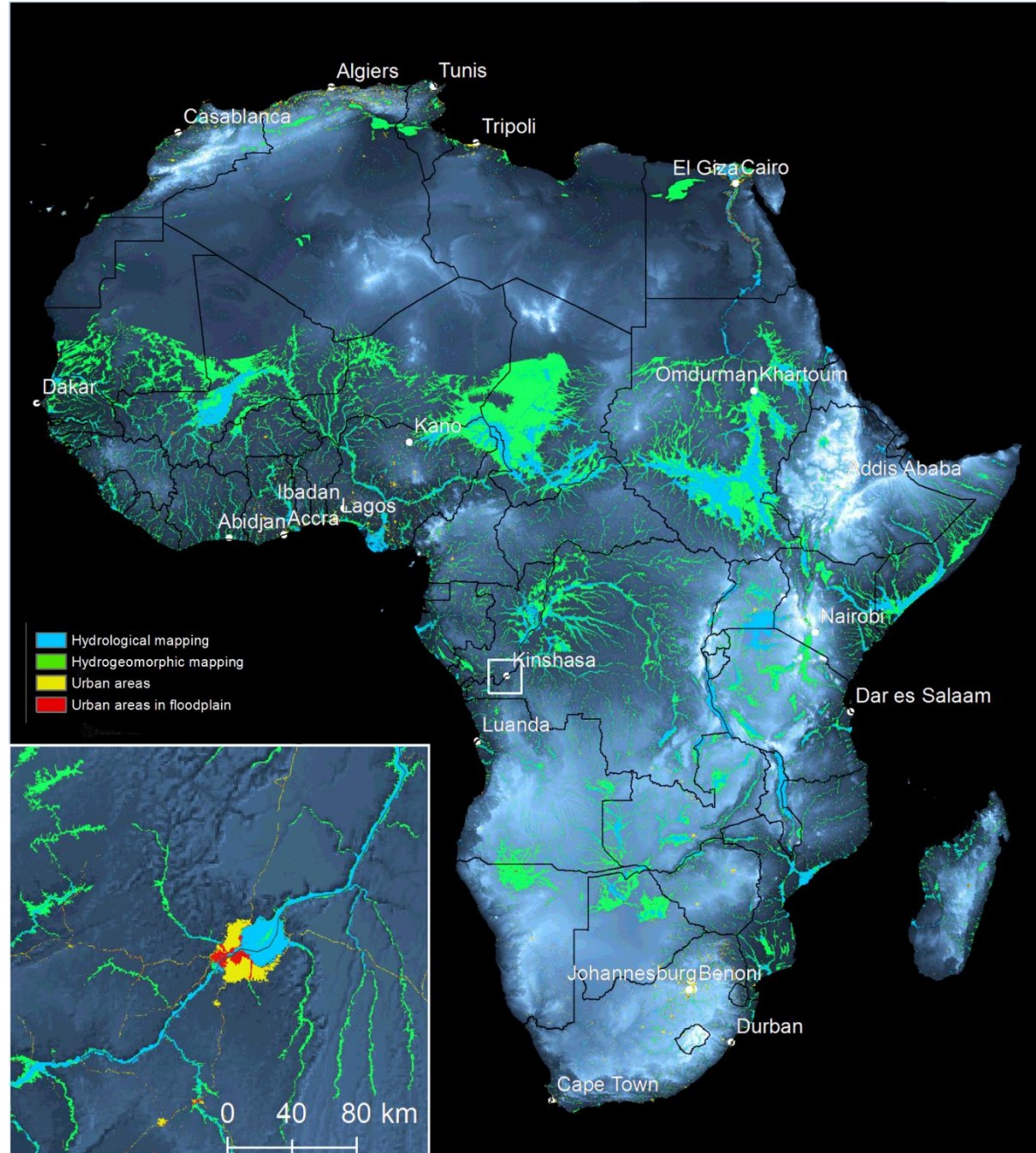


**Figure 1.** Hydrological vs. hydromorphic flood mapping in Africa. Continental floodplain mapping using an hydrological approach (in blue) with a return period of 200 years (Dottori et al., 2016) . The floodplain areas derived with the hydrogeomorphic approach (in green) are based on the GFPLAIN250m dataset4 (Nardi et al, 2019). The inset shows estimated flood-prone areas in Kinshasa (Democratic Republic of the Congo) as well as the Global Man-made Impervious Surface (GMIS) layer (Brown de Colstoun et al., 2017) depicting urban areas (in yellow) and urban areas in floodplains (in red).


This synthetic event is then propagated along the river with hydrodynamic models to estimate the corresponding inundated areas. The hydrological paradigm has been widely used across multiple places and scales (Ward et al., 2015), including large-scale flood hazard modelling in data-poor regions in Africa (Figure 1). While flood inundation modelling has been successful in simulating historical events (Schumann et al., 2013)., large uncertainties come into play when used to simulate synthetic

events (Di Baldassarre, 2012). The estimation of a flood hydrograph with a given return period, for example, is extremely uncertain as time series of flood data are hardly ever available, especially in data-poor areas (Blöschl et al., 2013). Trigg et al. (2016) compared flood maps produced by six hydrological models in Africa and found more disagreement than agreement.

## 3 Hydrogeomorphic mapping

An alternative paradigm to map flooding is based on a definition of floodplains as distinguished landscape features that have

been historically shaped by the accumulated effects of floods of varying magnitudes, and their associated hydrogeomorphic processes (Nardi et al., 2006; Dodov and Foufoula-Georgiou, 2006). In this *hydrogeomorphic* paradigm, floodplains are identified directly from the topography (Nobre et al., 2011; Nardi et al., 2019), which is assumed to have been shaped by past flooding events, and building on the concept of fractal river basins (Rodríguez-Iturbe and Rinaldo, 2001) or hydrogeomorphic theories (Tarboton et al., 1988). This paradigm does not require the estimation of a synthetic flood hydrograph, and

consistently identify flood-prone areas across different places (Manfreda et al., 2014; Nardi et al., 2018; Annis et al., 2019). Also, with the recent development of global DTMs (Ward et al., 2015; Nardi et al., 2019) and EO-based cloud computing platforms (Pekel, et al., 2016), worldwide mapping of floodplain areas is a reality and these global maps can be derived in a standard PC with a single click and limited computation time. Hence, it allows to easily detect floodplains, and it is a useful tool for a variety of environmental and socio-economic analyses at large or global scale.

**4 Comparing hydrological and hydrogeomorphic mapping**

Figure 1 shows, as an example, floodplains of the African continent derived with both paradigms (Dottori et al., 2016; Nardi et al., 2019), while its insert compares them in the area around the city of Kinshasa, Democratic Republic of the Congo. International development banks, water sector organizations, national and international bodies mandated with disaster risk reduction, sustainable development and humanitarian response use these global maps in data-poor regions for mapping flood

risk hotspots (Ward et al., 2015). To provide guidance in using these global products, we list limitations and advantages of the products derived using the two main paradigms in Table 1. This comparison focuses on the use of these global maps for the identification of flood-prone areas in data-scarce regions. It should be noted that the above paradigms have other purposes than flood mapping. Hydrological mapping is often carried out in order to derive probabilistic hazard metrics for risk assessment (e.g. Winsemius et al., 2016), while hydrogeomorphic maps can be used to support studies of anthropogenic pressure on rivers,

such as floodplain connectivity, as well as human-flood interactions (Lindersson et al., 2020).

**Table 1.** Advantages and limitations of the two paradigms in mapping floodplain areas.

| | Cons | Pros | Links to an example of global datasets (references) |
|---|---|---|---|
| **Hydrological mapping** | More sensitive to data scarcity. Ttime series of flood data are only seldom available and often too short for a robust estimation of a design flood (Blöschl et al., 2013).<br><br>Computationally expensive.<br><br>Variable over time, e.g. any interventions would require and updating of the hydrodynamic model. | Less sensitive to DEM inaccuracies (Annis et al., 2019; Nardi et al., 2019).<br><br>Floodplains are defined based on a specific probability of occurrence: this allows cost-benefit analyses and risk assessment (Winsemius et al., 2016).<br><br>It can explicitly account for the role of hydraulic structures, e.g. flood gates, or climate change.<br><br>It provides additional variables, such as maximum flow depth, velocity and volume useful for some applications. | Flood Hazard Maps at European and Global Scale by the Joint Research Center (JRC)<br><br>https://data.jrc.ec.europa.eu/collection/id-0054<br><br>(Dottori et al., 2016) |
| **Hydrogeomorphic mapping** | More sensitive to DEM inaccuracies (Annis et al., 2019; Nardi et al., 2019).<br><br>Do not provide a specific probability of occurrence: cost-benefit analyses for the design of e.g. risk reduction measures are not possible.<br><br>It cannot account for the role of hydraulic structures, e.g. flood gates, or climate change.<br><br>Scaling laws have limitations in dry climates. | Less sensitive to data scarcity, as it does not require any time series.<br><br>Computationally efficient (Annis et al., 2019).<br><br>More consistent over time, e.g. floodplain is identified as if protection structures were not in place. This can be seen as an advantage as erring on the side of least consequences (and total protection is impossible anyway). | Global High-resolution Dataset of Earth's Floodplains (GFPLAIN250m)<br><br>https://figshare.com/articles/GFPLAIN250m/6665165/1<br><br>(Nardi et al., 2019) |

# 5 Conclusions

Both paradigms are based on consolidated theories, and they have opposite advantages and uncertainties (Table 1). Thus, we
argue that these maps are complementary and they should be exploited following the precautionary principle (Foster et al., 2000), which is an important component of much of the environmental legislation in the western world. The principle calls for erring on the side of least consequences. In this context, this means that the identification of flood prone-areas in data-poor regions should consider flood inundation areas derived by the two paradigms. The insert of Figure 1, for instance, highlights (in red) the urban areas falling within the hydrological and/or hydrogeomorphic flood map. While, for the sake of simplicity,
our example considered only two global maps, the precautionary principle calls for using all existing flood maps. In this case, the growing availability of EO data (Schumann et al., 2009; Pekel et al., 2016; Lindersson et al., 2020) offer a great potential to test several maps and identify the (most credible) ones that can then be used to estimate flood-prone areas in data-poor regions.

**Data availability**

Maps and data are available online, and can be accessed using the links provided in Table 1.

**Author contributions**

GDB, FN, and SG conceptualized the study. AA prepared the figure with the support of GDB, FN and SG. GDB wrote the original draft of the brief communication. FN, AA, VO, MR, and SG provided comments and reviewed the original draft.

**Competing interests**

The authors declare that they have no conflict of interest.

**Acknowledgements**

This work was developed within the activities of the Panta Rhei research initiative of the International Association of Hydrological Sciences (IAHS). GDB, MR and VO are supported by the European Research Council (ERC) within the project "HydroSocialExtremes: Uncovering the Mutual Shaping of Hydrological Extremes and Society", H2020 Excellent Science,

Consolidator Grant no. 761678.

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
