# Peer review of "Brief communication: Comparing hydrological and hydrogeomorphic paradigms for global flood hazard mapping"

_Natural Hazards and Earth System Sciences, 2019_

## Referee Comment (RC1) · Francesco Dottori (Referee) · 20 Feb 2020

This Brief Communication proposes an useful comparison of two main paradigms for global flood risk modelling, and it will probably raise the interest of researchers and practitioners working on the topic. The manuscript has a clear scope and it is generally well structured, event though some sections could be improved. At the same time, while I understand the constraints of the manuscript format, I believe that some statements and conclusions need a bit more discussion and should be supported with adequate references.

General comments

1) I'm not fully convinced by the proposal of using the terms "top-down" (TD from now on) and "bottom-up" (BU from now on) to define the two modelling approaches. Is it a novel definition by the Authors, or does it come from previous literature? I'm not in principle against such definitions, but I would like the Authors to bring more compelling reasons for their use. Considering how these two terms are used in other contexts (e.g. in decision-making processes), the analogy proposed seems in my opinion misleading. On the one hand, all methods that delineate floodplains based on topograhy still require assumptions and elaborations which need to be consistent everywhere (i.e. the "top" side). At the same time, results of TD models are heavily influenced by topography, perhaps more that by hydrology (i.e the "bottom" side). Therefore, why not use the definitions of "hydrological" and "hydrogeomorphic" paradigms respectively? This would clearly indicate the main driver of each approach and is possibily more consistent with the terminologiy adopted so far in literature.

2) The present descriptions of the two paradigms (Sections 2-3) are not balanced: BU description has a large number of references (possibly not all of them necessary), with some sentences not so relevant in the context of this manuscript (see some specific comments below). In addition, it does not mention any drawback, for TD methods there are some lines on the uncertainty of estimating synthetic flood events. Given that pros and cons of the two approaches are listed in Table 1, my suggestion is to remove them and tighten up the two descriptions.

Specific comments

- Please specify in the abstract and introduction that the paper is about inland flooding only (coastal flooding seem not to be considered here).

- abstract: "resulting in a plethora of freely available products". To my best knowledge, most global scale models, are not freely available at global scale (with the excepotion of the JRC and GFPLAIN models). Are there more BU or TU models applied at global (or at least continental) scale,? If not, please consider rephrasing this sentence.

- Surprisingly, climate change is not mentioned after the introduction, yet it is a crucial field of application for global TD models

- line 26-27: I would write "...a range of synthetic events" instead of "a synthetic event", given that all global models use a range of flood magnitudes to evaluate flood hazard.

- Figure 1. The inset is not clear, it is hard to understand whether urban areas overlap with flood maps. I would suggest to avoid the color scale for water depth and just represent flood extent, possibly with a different color for representing the intersection of flood maps with impervious areas. Also, please put a scale in the inset and add a reference for the GFPLAIN dataset in the caption.

- lines 31: "while hydrodynamic models have been successful in reproducing historical events..." the reference provided is for a local-scale model, which is not so relevant in thix context (it is well understood that a hydraulic model is fit-for-purpose when applied at local scale). I would rather refer to results of large scale models such as in Schumann et al. (2013) and Wing et al (2019).

- line 49 "... identifies flood-prone areas across diverse climatic regimes with varying parameterizations." I think this sentence is not so relevant and could be removed. First, this is true not only for BU models but also for TD models. Second, it seems to partially contradict the statement in Table 1 that "Scaling laws have limitations in dry climates". Looking at Figure 1, my impression is that GFPLAIN is identifying ancient floodplains which are not anymore flood-prone areas, therefore suggesting that topography is not representative of present conditions. Maybe the Authors could add a reference to previous studies describing the performance of BU models in arid and semi-arid climates

- line 59-61 "International development banks (...) humanitarian response use these global maps in data-poor regions for mapping risk hotspots and flood-prone areas (Ward et al., 2015)". To my best knowledge, the paper by Ward et al only makes reference to global TD models . Are there references about the use of BU models in these contexts?

- Table 1: Please add references for the suggested pros and cons, where possible

- Table 1: "variable over time" is listed as a cons for TD models. However, this is also a strong pros, because models can simulate changing conditions (e.g. climate change scenarios, change in hydraulic structures...).

- Table 1: besides the design of risk reduction measures, TD models can be used for risk assessment in general.

- Table 1: I'm wondering whether BU models can somehow estimate flood magnitude. In my understanding, BU models identify a sort of residual risk, i.e. they map all the areas which may be affected by flooding, but withot an explicit representation of hazard frequency and magnitude. I woyld be happy to hear the opinion of the Authors on this point.

- Table 1: "more sensitive to scales": this is not much clear, do you mean that BU approaches have low skill over the minor drainage network? In my view, an advantage of BU methods is their feasibility of application over very fine scales (e.g. minor branches of any river network), especially conidering that the vast majority of global models are limited to major rivers because of data and computational constraints. If it's not the case, please add some further explanation or references.

Conclusions: "flood risk hotspots" should be changed to "flood prone areas" or similar definition, as according to the authors themselves BU models cannot quantify risk

Conclusions: "In this context, this means the identification of flood risk hotspots in data-poor areas should consider both flood inundation areas derived by the two paradigms as depicted in the insert of Figure 1." I agree in principle with such statement, which is in line the idea of using ensemble of models to better represent (at least in part) the overall uncertainty of model estimated. However, such conclusion should be better framed considering the existing literature. I'm listing some recent works below, my suggestion is to incorporate at least some of them (and possibly others, if manuscript

constraints allow) in the elaboration of conclusions:

- Trigg et al. (2016) compared flood maps produced by six different TD models in Africa and found more disagreement than agreement, thus suggesting large and mostly unexplained uncertainty in TD model structures.

- Wing et al., (2017) observed that global TD models have rarely undergone testing against high-quality data of commensurate coverage

- Bernhofen et al. (2018) observed that ensembles based on a range TD models had skill in reproducing three past flood events (but adding all models increased noise without increasing accuracy).

- Wing et al. (2019) evaluated the flood maps produced by large scale TD model against a BU model (HAND tool , NOAA 2018) and found significant lower performances by the BU model.

Conclusions: a personal opinion on the precautionary principle (up to the authors whether to use it or not): in my view, the precautionary principle should call for using all existing models, rather than one sample from each of the two paradigms. This is of course valid in an ideal world of free and open datasets, whereas in the real world the selection of global flood maps is in fact limited by data availability.

References

- Wing, et al. (2017) Validation of a 30 m resolution flood hazard model of the conterminous United States. Water Resources Research https://doi.org/10.1002/2017WR020917

- Trigg et al (2016) The credibility challenge for global fluvial flood risk analysis Environ. Res. Lett. 11 10

-Bernhofen et al (2018): A first collective validation of global fluvial flood models for major floods in Nigeria and Mozambique. Environmental Research Letters 13.10 104007.

- Wing et al (2019) flood inundation forecast of Hurricane Harvey using a continental-scale 2D hydrodynamic model. Journal of Hydrology X 4, 100039

- Schumann et al , 2013 A first large-scale flood inundation forecasting model, Water Resources Research, https://doi.org/10.1002/wrcr.20521

- NOAA National Water Center, Boghici, E., Arctur, D., 2018, NOAA NWC – Harvey NWM-HAND Flood Extents (HydroShare). doi:10.4211/hs.fe85a680d0144e79b39e8c483dc1e5aa.

---

## Referee Comment (RC2) · Guy J.-P. Schumann (Referee) · 6 Mar 2020

This piece of communication is well written and brings up an important point of discussion, that of advantages and disadvantages of floodplain delineation vs flood hazard return period computations. I took the liberty of reading Francesco Dottori's comments after reading the piece and before writing this since I think he raised very good points that I don't need to necessarily repeat but had several of the same.

I add here some other, more general thoughts. Although I think it is important to discuss the value of floodplain delineation, I think there are some considerable differences of using a hydrodynamic hazard model approach vs a hydro-geomorphic approach. I

think the method but above all the purpose and value of the approaches are very different and I don't think they can be so easily compared, but it is nonetheless useful to discuss these differences. I suggest the authors add a paragraph on this matter.

In fact, building a hazard model, the ultimate goal is to infer probabilistic hazard and then risk by looking at exposure and vulnerabilities. The Dottori model described in the paper does this first part (probabilistic hazard) and can be used to fuse with eventually the other parts of the risk equation and loss functions etc. These models, when set up globally, can also be run in stochastic mode and uncertainties can be estimated etc. Moreover, many have successfully shown that these models, at global level, or not as data-hungry as traditional reach scale CFD models or the like.

Using the hydro-geomorphic approach, however, it would be non-trivial to derive probabilistic hazard metrics - in fact I find it difficult to conceptualize even. It may be possible in some places around the globe, when very good topographic data are available and can depict floodplain extent changes like riverine terraces at different levels but, esp. given the very low accuracy freely available DEM data, I think this is not possible for many places.

I also think the aim of running a hazard model for floods is not to bound the floodplain but rather it is to understand the likelihood of a given flood magnitude returning and its effect in terms of hazard, such as inundation depth and extent. Also, these models can be used for off-main floodplain hazard computations like pluvial hazard or small floodplains traditionally not included in standard protection plans. Also, having built a model allows one to run it in inundation hindcast, nowcast and forecast as some have successfully illustrated. All this cannot be done with the hydro-geomorphic approach.

Also, the authors mention in the BU approach EO data but only briefly. Now this is interesting since they could have argued that EO of floods with a long history available, at least up to 1:20 if not 1:40 year floodplain inundation hazard (given 45 years of EO sensor history) can be estimated. But the authors did unfortunately not go any deeper

into this, which in my mind would have been very interesting. Moreover, EO can capture pluvial, coastal and riverine flooding which is difficult to separate out of course but at least it will capture those, which will be laborious to represent in flood hazard models accurately.

I think I put enough of my thoughts up and with Francesco's comments this should be enough food for thought. Nonetheless, as I said, this is a much needed and welcomed discussion.

---

## Author Comment (AC1) · 18 Mar 2020

We thank Francesco Dottori for this outstanding review. His constructive comments will help us improve the critical analysis of alternative paradigms for flood mapping.

We aim to substantially revise our brief communication. We agree with the Referees' main points:

1) Indeed, top-down and bottom-up can potentially be misleading as used different in other contexts. As suggested, we will use more common terms, i.e. hydrological vs. hydrogeomorphic, in the revised manuscript.

[Figure]

2) We will revise the manuscript by balancing the descriptions of the two paradigms and using a similar number of references. The ones that are not necessary will be removed.

Moreover, all specific comments will be carefully addressed in the revised manuscript and, in particular, the Figure will be revised as suggested (see revised figure below).
* * *
Legend:
- Hydrological mapping
- Hydrogeomorphic mapping
- Urban areas
- Urban areas in floodplain

Cities labeled: Algiers, Tunis, Casablanca, Tripoli, El Giza, Cairo, Dakar, Kano, Omdurman, Khartoum, Ibadan, Lagos, Abidjan, Accra, Addis Ababa, Nairobi, Kinshasa, Luanda, Dar es Salaam, Johannesburg, Benoni, Durban, Cape Town

0    40    80 km

**Fig. 1.**

---

## Author Comment (AC2) · 18 Mar 2020

We thank Guy Schumann for this outstanding review. His constructive comments will help us improve the critical analysis of alternative paradigms for flood mapping.

We aim to substantially revise our brief communication, revised its main figure (see below), address all comments (see also reponse to Dottori's comments) and we totally agree with the Referees' main points:

1) Indeed, our brief communication is lacking a description of the potentials and limitations of hydrological vs. hydrogeomorphic methods to make probabilistic flood maps.

This aspect will be included in the revised manuscript.

2) We will extend the part related to EO of floods. We recognize that the original manuscript did not fully recognize its long history. We will add text and references (within the limits allowed for a brief communication) to cover this additional dimension.

[Figure]

**Fig. 1.**

---

## Author Response (AR1)

**Response Letter**

**& Marked Manuscript (with track changes)**

**Referee #1**

This Brief Communication proposes an useful comparison of two main paradigms for global flood risk modelling, and it will

5 probably raise the interest of researchers and practitioners working on the topic. The manuscript has a clear scope and it is generally well structured, event though some sections could be improved. At the same time, while I understand the constraints of the manuscript format, I believe that some statements and conclusions need a bit more discussion and should be supported with adequate references.

We thank Francesco Dottori for this outstanding review. His constructive comments helped us improve the critical analysis of

alternative paradigms for flood mapping. We substantially revised our brief communication and address all Referees' points 10 (see point-by-point response and marked manuscript below).

**MAIN COMMENTS**

30

1) I'm not fully convinced by the proposal of using the terms "top-down" (TD from now on) and "bottom-up" (BU from now

- on) to define the two modelling approaches. Is it a novel definition by the Authors, or does it come from previous literature? 15 I'm not in principle against such definitions, but I would like the Authors to bring more compelling reasons for their use. Considering how these two terms are used in other contexts (e.g. in decision-making processes), the analogy proposed seems in my opinion misleading. On the one hand, all methods that delineate floodplains based on topography still require assumptions and elaborations which need to be consistent everywhere (i.e. the "top" side). At the same time, results of TD models are
- heavily influenced by topography, perhaps more that by hydrology (i.e the "bottom" side). Therefore, why not use the 20 definitions of "hydrological" and "hydrogeomorphic" paradigms respectively? This would clearly indicate the main driver of each approach and is possibily more consistent with the terminologiy adopted so far in literature

1) The Referee argues that the use of terms like top-down and bottom-up can potentially be misleading as used differently in other contexts. We agree and thus we used more common terms, i.e. hydrological vs. hydrogeomorphic (as suggested by the Referee). See revised manuscript with track changes, as well as revised Figure 1. 25

- The present descriptions of the two paradigms (Sections 2-3) are not balanced: BU description has a large number of references (possibly not all of them necessary), with some sentences not so relevant in the context of this manuscript (see some specific comments below). In addition, it does not mention any drawback, for TD methods there are some lines on the uncertainty of estimating synthetic flood events. Given that pros and cons of the two approaches are listed in Table 1, my
- suggestion is to remove them and tighten up the two descriptions. 2) The Referee argues that the descriptions of the two paradigms and the number of references was not balanced in the original

manuscript. We agree and thus we rephrased the description by addressing the specific comments below and removed

unnecessary references (as suggested by the Reviewer) for the hydrogeomorphic approach. The revised manuscript has the same number of references for the two paradigms.

35

**SPECIFIC COMMENTS**

Please specify in the abstract and introduction that the paper is about inland flooding only (coastal flooding seem not to be considered here).

Amended. Reference to river flooding was made explicit in the abstract and introduction in the revised manuscript.

40 abstract: "resulting in a plethora of freely available products". To my best knowledge, most global scale models, are not freely available at global scale (with the exception of the JRC and GFPLAIN models). Are there more BU or TU models applied at global (or at least continental) scale,? If not, please consider rephrasing this sentence.

Right. Freely available was removed.

- Surprisingly, climate change is not mentioned after the introduction, yet it is a crucial field of application for global TD

45 models

Indeed, hydrological approaches can help develop scenarios of climate change, which is not easy with hydrogeomorphic methods. We added this aspect in the revised Table 1.

- line 26-27: I would write "...a range of synthetic events" instead of "a synthetic event", given that all global models use a range of flood magnitudes to evaluate flood hazard.

50 Yes, amended.

Figure 1. The inset is not clear, it is hard to understand whether urban areas overlap with flood maps. I would suggest to avoid the color scale for water depth and just represent flood extent, possibly with a different color for representing the intersection of flood maps with impervious areas. Also, please put a scale in the inset and add a reference for the GFPLAIN dataset in the caption.

55 Thanks again. Yes, we agree. The figure was revised as suggested.

- lines 31: "while hydrodynamic models have been successful in reproducing historical events..." the reference provided is for a local-scale model, which is not so relevant in thix context (it is well understood that a hydraulic model is fit-for-purpose when applied at local scale). I would rather refer to results of large scale models such as in Schumann et al. (2013) and Wing et al (2019).

60 Good point. We used the suggested references instead.

- line 49 "... identifies flood-prone areas across diverse climatic regimes with varying parameterizations." I think this sentence is not so relevant and could be removed. First, this is true not only for BU models but also for TD models. Second, it seems to partially contradict the statement in Table 1 that "Scaling laws have limitations in dry climates". Looking at Figure 1, my impression is that GFPLAIN is identifying ancient floodplains which are not anymore flood-prone areas, therefore suggesting

65 that topography is not representative of present conditions. Maybe the Authors could add a reference to previous studies describing the performance of BU models in arid and semi-arid climates

The performance of hydrogeomorphic methods in arid and semi-arid climates is still to be tested. As such, we removed the sentenced in the revised manuscript.

- line 59-61 "International development banks (...) humanitarian response use these global maps in data-poor regions for

70 mapping risk hotspots and flood-prone areas (Ward et al., 2015)". To my best knowledge, the paper by Ward et al only makes reference to global TD models. Are there references about the use of BU models in these contexts? Only grey literature or informal information, but this is exactly one of the main objectives of our brief communication. Hydrogeomorphic theories are sounded, and thus they should be considered in flood mapping in data-scarce regions to

complement hydrological mapping (re: precautionary principle).

75 *Table 1: Please add references for the suggested pros and cons, where possible* Amended.

- Table 1: "variable over time" is listed as a cons for TD models. However, this is also a strong pros, because models can simulate changing conditions (e.g. climate change scenarios, change in hydraulic structures...).

In fact, it was also listed as a pros in Table 1, which now states "It can explicitly account for the role of hydraulic structures, 80 e.g. flood gates, or climate change."

- *Table 1: besides the design of risk reduction measures, TD models can be used for risk assessment in general.* Yes, added to the revised Table 1.

- Table 1: I'm wondering whether BU models can somehow estimate flood magnitude. In my understanding, BU models identify a sort of residual risk, i.e. they map all the areas which may be affected by flooding, but withot an explicit representation of

85 hazard frequency and magnitude. I woyld be happy to hear the opinion of the Authors on this point. The Referee is right. The hydrogeomorphic maps do not make an explicit representation of hazard frequency and magnitude. This is stated in the Table 1 as pros for hydrological mapping and cons for hydrogeomorphic methods. Moreover, we added a paragraph in the conclusions to clarify this difference (see also our response to Referee #2).

Table 1: "more sensitive to scales": this is not much clear, do you mean that BU approaches have low skill over the minor
drainage network? In my view, an advantage of BU methods is their feasibility of application over very fine scales (e.g. minor
branches of any river network), especially conidering that the vast majority of global models are limited to major rivers
because of data and computational constraints. If it's not the case, please add some further explanation or references.

We agree with Referee that "more sensitive to scales" is misleading. Indeed, hydrogeomorphic approaches are able to capture floodplain extents at finer scales covering also minor upstream tributaries, while global applications of hydrodynamic models

- 95 are usually not available. We used the term "scales" here to refer to DEM resolution and accuracy. As tested in Annis et al. (2019) and Nardi et al. (2019), the performance of hydrogeomorphic methods changes extensively with DEM resolution and source/accuracy. Hydrodynamic models are also highly impacted by DEM resolution and accuracy, but less than hydrogeomorphic models, as the former approach mainly evaluates elevation differences between fluvial channel and floodplain areas, while the latter also includes governing hydrologic dynamics of inundation processes (flood volume in
- 100 particular) that diminish the impact on results of DEM inaccuracies. As a result, hydrogeomorphic methods suffer more than

hydrological approach when complex hydrologic dynamics and fluvial landscape features are governing factors of inundation processes/extents. Accordingly, we revised Table 1 that now reads as "more sensitive to DEM inaccuracies (Annis et al., 2019; Nardi et al., 2019)."

Conclusions: "flood risk hotspots" should be changed to "flood prone areas" or similar definition, as according to the authors themselves BU models cannot quantify risk

**Amended.**

105

Conclusions: "In this context, this means the identification of flood risk hotspots in datapoor areas should consider both flood inundation areas derived by the two paradigms as depicted in the insert of Figure 1." I agree in principle with such statement, which is in line the idea of using ensemble of models to better represent (at least in part) the overall uncertainty of model

110 estimated. However, such conclusion should be better framed considering the existing literature. I'm listing some recent works below, my suggestion is to incorporate at least some of them

We thank the Referee for suggesting more references. They were all considered in reviewing the paper. Schumann et al. (2013) and Trigg et al. (2016) were included as new references. They are very relevant for our brief communication. Yet, while we found the papers by Bernhofen and Wigg are excellent, we did not include them because of their focus on the reproduction of

115 specific events.

Conclusions: a personal opinion on the precautionary principle (up to the authors whether to use it or not): in my view, the precautionary principle should call for using all existing models, rather than one sample from each of the two paradigms. This is of course valid in an ideal world of free and open datasets, whereas in the real world the selection of global flood maps is in fact limited by data availability.

120 We agree with the Referee that, in an ideal world where all datasets were freely available, one could use ensemble of all existing models. Hence, we rephrased the conclusions to open up for such an opportunity and clarified that we picked up only two maps as examples here for the sake of simplicity. See revised manuscript.

**Referee #2**

125 This piece of communication is well written and brings up an important point of discussion, that of advantages and disadvantages of floodplain delineation vs flood hazard return period computations. I took the liberty of reading Francesco Dottori's comments after reading the piece and before writing this since I think he raised very good points that I don't need to necessarily repeat but had several of the same.

We thank Guy Schumann for this outstanding review. His constructive comments helped us improve the critical analysis of

130 alternative paradigms for flood mapping. We substantially revised our brief communication. (see point-by-point response and marked manuscript below). I add here some other, more general thoughts. Although I think it is important to discuss the value of floodplain delineation, I think there are some considerable differences of using a hydrodynamic hazard model approach vs a hydro-geomorphic

- 135 approach. I C1 NHESSD Interactive comment Printer-friendly version Discussion paper think the method but above all the purpose and value of the approaches are very different and I don't think they can be so easily compared, but it is nonetheless useful to discuss these differences. I suggest the authors add a paragraph on this matter. Using the hydro-geomorphic approach, however, it would be non-trivial to derive probabilistic hazard metrics - in fact I find it difficult to conceptualize even. It may be possible in some places around the globe, when very good topographic data are available and can depict
- 140 floodplain extent changes like riverine terraces at different levels but, esp. given the very low accuracy freely available DEM data, I think this is not possible for many places. I also think the aim of running a hazard model for floods is not to bound the floodplain but rather it is to understand the likelihood of a given flood magnitude returning and its effect in terms of hazard, such as inundation depth and extent. Also, these models can be used for off-main floodplain hazard computations like pluvial hazard or small floodplains traditionally not included in standard protection plans. Also, having built a model allows one to
- 145 run it in inundation hindcast, nowcast and forecast as some have successfully illustrated. All this cannot be done with the hydro-geomorphic approach.

We agree with the Referee. Our brief communication is lacking a description of the different purposes of hydrological vs. hydrogeomorphic methods and an explicit reference to probabilistic flood maps. This aspect has been included in the conclusions of the revised manuscript. In particular, we clarify that our comparison focuses on the use of these global maps

150 for the identification of flood-prone areas in data-scarce regions. We also noted that the above paradigms have other purposes than flood mapping. Hydrological mapping is often carried out in order to derive probabilistic hazard metrics for risk assessment, while hydrogeomorphic maps can be used to support large-scale studies of human-flood interactions or anthropogenic pressure (e.g. floodplain dysconnectivity) on rivers.

Also, the authors mention in the BU approach EO data but only briefly. Now this is interesting since they could have argued

- 155 that EO of floods with a long history available, at least up to 1:20 if not 1:40 year floodplain inundation hazard (given 45 years of EO sensor history) can be estimated. But the authors did unfortunately not go any deeper C2 NHESSD Interactive comment Printer-friendly version Discussion paper into this, which in my mind would have been very interesting. Moreover, EO can capture pluvial, coastal and riverine flooding which is difficult to separate out of course but at least it will capture those, which will be laborious to represent in flood hazard models accurately.
- 160 2) We extended the part related to EO of floods. We added text and references (within the limits allowed for a brief communication) to cover this additional dimension. We think that the growing availability of EO data offer a great potential to test several maps and identify the (most credible) ones that can then be used to estimate flood-prone areas in data-poor regions.

**Brief communication: Comparing top-downhydrological and bottom-170 uphydrogeomorphic paradigms for global flood hazard mapping**

Giuliano Di Baldassarre1,2,3, Fernando Nardi4,5, Antonio Annis4, Vincent Odongo1,3, Maria Rusca1,3, and Salvatore Grimaldi6

1Department of Earth Sciences, Uppsala University, Uppsala, Sweden 2IHE-Delft Institute for Water Education, Delft, The Netherlands

3Centre of Natural Hazards and Disaster Science, CNDS, Sweden
 4WARREDOC, University for Foreigners of Perugia, Perugia, Italy
 5Institute of Water & Environment, Florida International University, Miami, USA
 6Tuscia University, Viterbo, Italy

Correspondence to: Giuliano Di Baldassarre (giuliano.dibaldassarre@geo.uu.se)

180 **Abstract.** Global floodplain mapping has rapidly progressed over the past few years. Different methods have been proposed to identify areas prone to river flooding, resulting into a plethora of freely available products. Here we assess the potential and limitations of two main paradigms, and provide guidance on the use of these global products in assessing flood risk in data-poor regions.

**1** Introduction**

- As economic losses and fatalities caused by river floodings have dramatically increased over the past decades (Winsemius et al., 2016), there has been much progress in the development of analytical tools for the identification of the areas that can be potentially flooded (Ward et al., 2015; Dottori et al., 2018; Nardi et al., 2019). This progress has also been accelerated by the adoption of the Sendai Framework for Disaster Risk Reduction and the Warsaw International Mechanism for Loss and Damage Associated with Climate Change Impacts (Ward et al., 2015). As such, more and more scientists, experts and practitioners use
- 190 global floodplain maps in data-poor regions for the identification of flood risk hotspots or the mapping of flood-prone areas (Ward et al., 2015; Winsemius et al., 2016; Dottori et al., 2018; Nardi et al., 2019).

**2 The top-downHydrological mapping paradigm**

There are two main paradigms to map flooding. The traditional paradigm is (implicitly or explicitly) based on a definition of the floodplain as the area falling within the extent of a given flood event. In this hydrological paradigm, <del>which can be seen as</del>

- 195 top-down, a range of synthetic event with a given probability of occurrence or return period (Pappenberger et al., 2013; Ward et al., 2015; Dottori et al., 2018), such as the 1-in-200 year flood event, is typically estimated via hydrological modelling or statistical analysis of flood data. This synthetic event is then propagated along the river with hydrodynamic models to estimate the corresponding inundated areas. The top-downhydrological paradigm has been widely used across multiple places and scales (Ward et al., 2015), including large-scale flood hazard modelling in data-poor regions in Africa (Figure 1). While
- 200 hydrodynamic-flood inundation modelling of floods-has been successful in simulating historical events (Schumann et al., 2013).Horritt and Bates, 2002), large uncertainties come into play when used to simulate synthetic events (Di Baldassarre, 2012). The estimation of a flood hydrograph with a given return period, for example, is extremely uncertain as time series of flood data are hardly ever available, especially in data-poor areas (Blöschl et al., 2013). Trigg et al. (2016) compared flood maps produced by six hydrological models in Africa and found more disagreement than agreement.